# Clean Household Energy Consumption and Residents’ Well-Being: Empirical Analysis and Mechanism Test

**DOI:** 10.3390/ijerph192114057

**Published:** 2022-10-28

**Authors:** Pengyu Ren, Xiaoyi Liu, Fanghua Li, Dungang Zang

**Affiliations:** College of Economics, Sichuan Agricultural University, Chengdu 611130, China

**Keywords:** clean household energy, residents’ well-being, CHARLS, IV-O-Probit model, CMP method

## Abstract

Improving well-being is a critical problem for worldwide social progress. Research on well-being effects of clean household energy consumption is of great significance for the realization of United Nation’s Sustainable Development Goals (SDGs). Due to the multifaceted role of clean household energy in enhancing well-being as a sustainable development goal, it has attracted extensive academic attention and research but still needs to be further refined through new perspectives. This paper uses data from the 2018 China Health and Retirement Longitudinal Study to conduct an empirical analysis of clean household energy consumption and residents’ well-being using an ordered probit model, the instrumental variables method, a conditional mixed process (CMP) method, and a mechanism analysis model. The results show that (1) household clean energy consumption contributes to residents’ well-being, and the results remained significant after selecting “Do you have an electric Bicycle?” as an instrumental variable to address the endogeneity question. (2) According to heterogeneity research, women’s wellbeing is impacted by clean energy consumption in the household more than men’s. Only in rural locations can clean household energy consumption significantly boost well-being; urban and suburban areas are unaffected. (3) In the mechanism analysis, health condition and depression played a mediating role on the impact of clean household energy consumption on well-being, and social contact played a moderating role on the impact of clean household energy consumption on well-being. The findings and policy recommendations in this paper are suggestive of how we can improve the well-being of residents in low- and middle-income countries and provide reference values for research in related fields around the world.

## 1. Introduction

The United Nations (UN) identified “ensuring access to affordable, reliable, and sustainable modern energy for all” as one of its goals to transform the world. Clean energy is essential for improving people’s well-being and promoting social progress. Household energy use accounts for a large proportion of global final energy consumption [1]. Therefore, the development and promotion of household clean cooking technology is of great significance for improving the health and well-being of citizens so as to accomplish global Sustainable Development Goals (SDCs) [2].

The use of coal, wood, and other non-clean fuels will reduce residents’ life satisfaction by affecting their physical health [3]. Atmospheric pollution from combustion, such as PM2.5 and sulfur dioxide, increases the likelihood that groups exposed to high pollution environments and poor health conditions will fall into the environment–health–poverty trap [4] and reduce their personal well-being [5,6]. Energy poverty usually refers to the lack of the ability to pay or access to basic energy services to meet the most common needs of life [7], mainly in the form of the simultaneous denial of access to and affordability of clean energy [8]. One of the key reasons for energy poverty is the lack of access to modern energy sources that are relatively cleaner than traditionally used energy sources [9], such as clean cooking fuels [10]. Moreover, energy poverty has a negative impact on subjective well-being, which is mainly manifested by affecting health [11], interfering with social equity [12], and reducing food expenditure [13]. Evidence from Europe suggests that energy scarcity will lead to consumer vulnerability [14]. The EU and its member states have made significant progress in developing a strong policy-making framework to alleviate energy poverty [15], and different regions should take different measures to address energy poverty based on local realities [16]. Access to sustainable and affordable clean energy is an important component of energy security. Energy prices have a statistically and economically significant negative impact on subjective well-being [17], and price increases will significantly reduce the well-being of households in need and low-income households [18]. Focusing on households with high fuel poverty rates will be more effective in compensating for increased individual well-being [19]. The transformation from non-clean energy use to clean energy is conducive to improving personal happiness [20] and the welfare level of residents and communities [21,22], which plays an important role in increasing social well-being.

Macroscopically, the use of clean energy can reduce carbon emissions to facilitate national and regional economic upgrading, enhance employment [23], and national public health security [24], as well as maintain energy economic security [25], which plays an irreplaceable role in enhancing social well-being. At the micro-level, the consumption of clean energy significantly raises personal well-being and life satisfaction [20]. Households with clean energy can save labor time and reduce drudgery [26] and promote female health by extending female socialization time [27]. The shift to regularization of clean cooking fuels can reduce air pollution and thus bring health benefits [28]. The degree of air pollution decreases, and residents’ personal satisfaction significantly improves [29]. At the same time, residential electricity consumption can improve subjective well-being, narrow the difference between men and women, and improve the problem of educational imbalances [30].

Globally, “Household Clean Energy Use” has become a trending topic in energy transition research [31], focusing on the analysis of household energy choices and the well-being of clean energy transition. Data from rural Ghana show that clean energy consumption increases the proportion of healthy household members, with a more pronounced impact on women’s health [32]. In India, the transition to clean energy greatly improved the harm caused by PM2.5 pollution, and when everyone achieves a complete transition to clean energy, huge public health benefits can be achieved [24]. Beyond physical and mental health and environmental benefits, Aydin, E. et al. used Turkey as an example to examine the economic benefits of household clean energy in emerging economies; that is, reducing total household energy consumption while increasing the value of home ownership and rental housing [33]. Jan et al. assessed the key benefits of household energy transitions in Poland by proposing the concept of “everyday life” co-benefits to refer to the positive effects on well-being [1]. Ma et al. used the 2016 China Labor Dynamics Survey (CLDS) to study the impact of home cooking fuel choices on individual subjective well-being and found that complete energy switching significantly improved individual well-being and life satisfaction [20]. However, in fact, for low- and middle-income countries, research on the well-being of clean energy adoption at the household level is still lacking [34].

The current literature suggests that the long-term household use of non-clean energy reduces residential well-being and that a shift from non-clean to clean energy sources can increase residential well-being in multiple ways. However, we found limitations in existing relevant studies: (1) Existing studies are mostly from South Asian countries such as India and Bhutan, and there are relatively few microdata studies from China, a major energy-using country. (2) Most studies on well-being are related to health, income, etc., and a few have been conducted on energy poverty and air quality, while little attention has been paid to the deeper issues of the impact of clean energy use in households on well-being. (3) Many documents neglect the possible endogeneity of models. (4) Studies on the effect of clean household energy on well-being have mostly conducted single-conclusion analyses, without in-depth research on the mechanisms between them. (5) Most studies have not conducted heterogeneity analyses on the impact of clean household energy on well-being and have not considered whether there are different conclusions among different groups.

The purpose of this paper is to empirically test clean household energy and well-being by using the 2018 China Health and Retirement Longitudinal Study (CHARLS) and to conduct a deeper analysis of the underlying mechanisms. The results can provide new perspectives on China for related studies in the world, enrich the research content in related fields, and provide references for future studies. The innovative points of this study include the following: (1) CHARLS is China’s largest source of high-quality micro-data on households and individuals, opening up new avenues for clean energy and well-being research; (2) conducting a deep-dive study on the impact of clean household energy consumption on well-being and conducting robustness tests by replacing the model and replacing the sample’s data; using the instrumental variables method and the conditional mixed process estimation method to solve potential endogenous problems; (3) using mediating effects and moderating models to uncover possible mechanisms between clean household energy consumption and well-being, the empirical results were tested for heterogeneity by gender and region. 

The remaining sections include Data and Methods (Section 2), Empirical Analysis and Discussion (Section 3), Mechanism Analysis (Section 4), Discussion (Section 5), Conclusions and Policy Recommendations (Section 6).

## 2. Data and Method

### 2.1. Data

The data used in this study come from the China Health and Retirement Longitudinal Survey (CHARLS), which is a set of high-quality micro-data representing families and individuals of middle-aged and elderly people aged 45 and above in China. The content of the CHARLS questionnaire includes basic personal information of respondents, family structure and financial support, health status, anthropometric measurements, medical service utilization and medical insurance, work, retirement and pensions, income, consumption, assets, and basic community conditions, etc. Moreover, the CHARLS survey, which is representative at the national level, covered 28 provinces of China, 150 countries or districts, and 450 villages or urban communities across the country. Specifically, this study selected the 2018 national tracking survey, which was officially released in September 2020. It has high timeliness and can meet the needs of empirical research and mechanism analyses in this paper. The missing values of key variables were processed by means of mean filling, median filling, and elimination. That is to say that some samples with missing values are deleted and missing values of the corresponding variables are filled in with the mean or median of the existing data. Then, 19,683 final samples were obtained via matching processing.

### 2.2. Variables

#### 2.2.1. Response Variable: Residents’ Well-Being (WB)

Generally speaking, residents’ well-being reflects the satisfaction and security of residents in a country or a certain region. It refers to the experience of people’s satisfaction with their own lives. However, considering that the definition and measurement methods of well-being have not been unified in the academic community [35], this study, based on the individual subjectivity of residents’ well-being and referring to measures of well-being of most scholars, finally takes “life satisfaction” as the alternative variable for measuring residents’ well-being. Next, we chose the question “Overall, are you satisfied with your life?” to measure it. On this basis, this paper assigns the original value of this variable in reverse order, which means that 5 represents “extremely satisfied,” and the smaller the value, the lower the well-being.

#### 2.2.2. Explanatory Variable: Clean Household Energy (CHE)

According to the questionnaire, “What is the main fuel used for cooking in your home?” We construct the 0–1 variable to represent clean household energy consumption. When respondents answered “Natural gas, biogas, liquefied petroleum gas, and electricity,” it indicated that the household used clean energy, which was assigned a value of 1. The results of the other responses were assigned a value of 0.

#### 2.2.3. Control Variables

In addition to the above variables, considering that residents’ happiness may be affected by multi-dimensional factors, in order to increase the explanatory power of the model, factors affecting residents’ well-being will be more comprehensive and complete. We selected a series of variables that may affect residents’ well-being as control variables at the individual level (such as gender, marriage, work, debt, etc.), at the household level (such as regions of residence), and at the social level (such as medical level). The specific selected variables are shown in Table 1.

### 2.3. Variable Descriptive Statistics

In this paper, descriptive statistics for each variable were performed using STATA software, as shown in the Table 2 below. The values taken for each indicator of the study results are within an acceptable range (mean, standard deviation, maximum value, and minimum value). Most respondents felt happy (57.67%), while only a small proportion (10.71%) thought they were unhappy or very unhappy, indicating that personal well-being gradually changed from unhappy to happy. More than half of the families will prefer clean fuel when cooking (69.73%), which shows the important role of clean fuel in daily life. Moreover, most families purchased electric bicycles (41.88%), indicating that the awareness of environmental protection has increased. Among the respondents, the number of women is more than that of men. Most of them are married (85.12%), and the educational background is generally low, as nearly half of them (43.23%) are illiterate or semi-illiterate, and most people are not working (71.64%). In terms of income and expenditure, respondents do not have a high level of consumption and almost no debt. Nearly three-quarters of respondents live in rural areas (71.21%) and are satisfied with local medical services (84.83%). Air satisfaction represents the air quality of a region, and about 84.55 percent of the respondents are satisfied with air quality, demonstrating that air quality has improved. The average happiness of the respondents was 20.27, general depression was low, the physical condition was good (75.88%), but they did not like carrying out social activities with friends (66.26%).

### 2.4. Models

For life satisfaction, the explained variable for measuring residents’ well-being, is an ordered discrete variable with values ranging from 1 to 5, and this paper uses the ordered probit model for empirical analysis. The model equations are as follows: Equations (1) and (2).
(1)$$WB_{i}^{*}=\varphi_{n}+\beta_{n}CHE+\delta_{n}CV_{r}+\mu_{k}$$
(2)$$WB^{*}=\{\begin{array}{c} 1\;\;\;\;\;if\;0<i\leq 1\\ 2\;\;\;\;if\;1<i\leq 2\\ 3\;\;\;\;if\;2<i\leq 3\\ 4\;\;\;\;if\;\;3<i\leq 4\\ 5\;\;\;\;if\;4<i\leq 5 \end{array}$$

The $WB^{*}$ is potential life satisfaction; CHE is clean household energy; WB=1,2,3,4,5 represents different self-evaluations of life satisfaction; $\varphi_{n}$ is the intercept term; $\beta_{n}$ and $\delta_{n}$ are regression coefficients; $CV_{r}$ is the control variable; $\mu_{k}$ denotes the error term.

In order to explore the possible mediating mechanism between clean household energy consumption and well-being, this paper set up a mediating effect model as follows by referring to the relevant literature (Equation (3)) [36].
(3)$$WB_{i}^{*}=\varphi_{n}+\beta_{n}CHE+\delta_{n}CV_{r}+\mu_{k}\;\;\;MV=\varphi_{1}+\beta_{1}CHE+\delta_{1}CV_{r}+\mu_{1}\;WB_{i}^{*}=\varphi_{2}+\beta_{2}CHE+\sigma MV+\delta_{2}CV_{r}+\mu_{2}$$

Among them, *MV* is the mediating variable, and σ is the regression coefficient of the mediating variable. If $\beta_{n}$*,*
$\beta_{1}$*,*
$\beta_{2}$, and σ are all significant, this indicates the existence of the mediating effect.

Moreover, in addition to the mediating effect, there may also be a moderating mechanism between clean household energy consumption and residents’ well-being. The moderating effect model is established as follows (Equation (4)):
(4)$$WB_{i}^{*}=\varphi_{n}+\beta_{n}CHE+\delta_{n}CV_{r}+\mu_{k}\;WB_{i}^{*}=\varphi_{3}+\rho RV+\delta_{3}CV_{r}+\mu_{3}\;WB_{i}^{*}=\varphi_{4}+\theta \left(CHE\times RV\right)+\delta_{4}CV_{r}+\mu_{4}$$

*RV* is the moderating variable, and ρ and θ are regression coefficients. If $\beta_{n}$*,*
ρ*,* and θ are all significant, this indicates that *MV* plays a moderating role in the relationship between the use of *CHE* and *WB*.

## 3. Empirical Analysis and Discussion

### 3.1. Basic Regression

Table 3 model (1) details the regression results of the ordered probit model of clean household energy consumption on well-being, and model (2) shows the regression results of the average marginal effect. The results of column (1) show that clean household energy consumption has a significant positive impact on residents’ well-being. (2) With respect to the column with well-being choices 1 and 5, for example, the probability of well-being choices 1 will decrease by 0.004 and the probability of well-being choices 5 will increase by 0.007 when clean household energy consumption increases by 1 unit. The trend of regression results indicates that the consumption of clean energy in the household favors the improvement of residents’ well-being.

In addition to the regression results of clean household energy consumption on residents’ well-being, Table 3 also reflects the influence of control variables on well-being. Gender is significantly and positively correlated with well-being, and an increase in the proportion of women can boost well-being, which may be related to the fact that men are given greater expectations and face greater work burden and pressure in society [37]. Marriage has a significant positive effect on well-being, indicating that those with a married status were happier than those with an unmarried status [38]. The effect of increasing family expenditure on well-being is consistent with existing research results [39], which is conducive to the improvement of well-being. On the contrary, family debt seriously impacts the improvement of well-being, and debt repayment will cause pressure and affect emotional and mental health [40]. In terms of medical treatments, the purchase of medical insurance can reduce the medical burden and improve the convenience of medical treatments [41], thus inducing a significant positive impact on well-being. The more satisfied people are with the medical level, the happier they will feel because it can contribute to the improvement of residents’ well-being in terms of adequacy, publicity, and convenience [42].

The highest level of education, participation in work, and type of region have no significant effect on well-being. The regression coefficient of participation in work (0.015) is positive, implying that employment may enhance personal happiness. The regression coefficient of education level (−0.009) and regions (−0.015) is negative, the effect on well-being may be related to a variety of comprehensive factors, and the results are not meaningful to this study, so we will not provide an in-depth discussion.

### 3.2. Robustness Check

In order to further test the robustness of the results, this paper utilizes two methods: replacing the empirical model and sampling data to conduct the robustness test. As shown in Table 4, columns (2) and (3) replace the ordered probit model with the OLS and ordered logit model, respectively. The coefficients of clean energy for households after replacement are all positive and have a significant effect on residents’ well-being, consistent with the results before replacement. In the average marginal effect trend in column (3), the probability of well-being being 1 decreases by 0.004 and the probability of well-being being 5 increases by 0.006 for every one-unit increase in clean household energy consumption, indicating that increasing clean household energy use can indeed improve residents’ well-being. The Chinese General Social Survey (CGSS) is the first nationwide, comprehensive, and continuous academic survey project in China. It has become the main data source for the study of Chinese society, among which the survey on energy consumption and social attitudes can provide real and accurate data for this study. The regression results of the three models above were analyzed using the CGSS sample’s data, and the results are shown in (4), (5), and (6) columns. The OLS regression results in column (5) show that clean household energy consumption is positively correlated with well-being at a significance level of 1%. The coefficients of the two columns (4) and (6) are positive and significant. Clean household energy can significantly improve residents’ well-being. Combining the two regressions average marginal utility results, they both demonstrate that clean household energy consumption can gradually improve well-being. These results confirm that clean household energy consumption has a reliable and robust effect on well-being.

### 3.3. Endogeneity: Instrumental Variables Approach and CMP Estimation (IV-O-Probit Model)

Considering endogeneity problems caused by the measurement error of variables, omitted variable bias, and sample selection bias, the regression estimation results are unreliable. This article uses “whether the respondents and their spouses have electric bicycles (EB, 1 = yes, 0 = no)” as an instrumental variable to overcome the endogeneity problem. Generally speaking, the question of whether or not to own an electric bicycle will affect the user’s energy decision. Electric bicycles use electricity as energy, so the users who buy them may be more environmentally conscious, and it is inferred that they will also consider energy saving and emission reduction in their home energy choices, so as to use more clean energy. However, owning an electric bicycle had no significant effect on personal well-being. Based on this, we initially believe that “whether there is an electric bicycle” is an effective instrumental variable in theory.

Since previous research studies use the O-Probit model, it is technically infeasible to directly use the instrumental variable method for the ordinal model and ordinal variables [43]. Therefore, to overcome the endogeneity problem of the model, we combine the instrumental variable and conditional mixture process’s (CMP) estimation method to perform weak instrumental variable detection and solve the endogeneity problem [44]. Models (1) and (2) in Table 5 report regression results of the 2SLS two-stage least squares method, in which the F-statistic of the first-stage regression is 533.31 > 10, so it passes the weak instrumental variable test. That is to say that “whether there are electric bicycles” and “clean energy” are strongly correlated. The statistic atanhrho_12 constructed by the CMP estimate is significantly different from 0 (*p*-value = 0), as shown in models (3) and (4) (in Table 5). Therefore, it is believed that the core explanatory variable “CHE” does have obvious endogeneity, and there is a significant correlation between the two equations in the joint equation model. The results of the joint equation model for estimating the conditional mixture process are more effective than single estimation. In addition, after endogeneity treatments, the positive effect of CHE on residents’ subjective well-being significantly improved from 0.0698 in the basic regression to 0.3135 now, and this effect is still significant. It is further explained that there is a negative bias in the basic regression that does not overcome endogeneity. By introducing instrumental variables, this study effectively eliminates the influence of endogeneity, making the empirical results more robust and reliable.

### 3.4. Heterogeneity Analysis

After the previous analysis, we can conclude that, in general, the household consumption of clean energy can improve the subjective well-being of users, but there is no specific analysis of the effect’s differences between different groups. Relevant research shows that women may have more opportunities to use energy and fuel in the household than men, and the consumption of clean energy has a more profound impact on women [32]. In addition, different household addresses may lead to differences in their endowment for clean energy use [45]. To this end, this study classifies the sample by gender and the region of residence of respondents and further examines the heterogeneity of CHE effects on WB.

From the perspective of gender, the results of models (1) and (2) in Table 6 show that women’s use of clean energy has a significantly higher effect on their happiness (0.0716) than men’s (0.0699). In fact, since most households are often women who undertake the daily housework, women are more likely to be harmed by unclean energy solid particle pollutants in the process of these labor activities, such as cooking [46]. At the same time, CHE allows women to save considerable hours of labor each day [47]. Therefore, via the use of clean energy, the female group improved physically and psychologically, which in turn enhances their well-being even more strongly.

Models (3), (4), and (5) (in Table 6) show the differences in the effects of different regions of residence of respondents. According to the questionnaire, we divide them into three types: urban areas, urban–rural combination areas, and rural areas. The results showed that CHE only significantly improved happiness in rural areas and had no significant effect on the other two types. The reason is that compared with rural areas, urban areas are often the first to use clean energy, and the coverage of clean energy in urban areas is higher than that in rural areas. Therefore, for most urban households, clean energy has become commonplace in household energy consumption, and the simple transition from solid fuels to clean energy will not significantly improve their well-being. However, in rural areas, since the popularization of clean energy is slower than in urban areas, they have not achieved a complete transformation of clean energy, and some residents still use solid fuels and clean energy in a mixed manner. According to this, there is still considerable room for improving the well-being of rural residents by consuming clean energy, and the happiness of rural residents will be significantly improved.

## 4. Mechanism Analysis

### 4.1. Testing for Mediating Effects of Health Condition (HC) and Depression

We further explore the impact mechanism of CHE on WB and explain how clean energy consumption affects residents’ well-being. This paper sets the respondent’s health condition (HC) and depression as mediating variables and tests whether there is a mediating effect by the three-step regression method, Sobel method, and Bootstrap method. Among them, the HC is obtained from the question “How do you think your health condition is?”, which is also assigned in reverse order; that is, 1 represents “very bad”, and the larger the value, the better the health condition. The degree of depression refers to the practice of Wang Qiong and Zeng Guoan [48]. That is, the 10 questions on the depression scale in the CHARLS questionnaire that reflect the degree of depression of the respondents are summed up, and each question has a value of 1–4, and the final value after the summation is 10–40. The larger the value, the higher the degree of depression.

The results of models (1), (2), and (3) in Table 7 show that clean household energy consumption significantly improves the health condition of users, and the better the HC, the higher the WB. It is preliminarily verified that CHE affects WB by affecting HC, which means that HC plays a mediating effect on CHE impact on WB. Consuming clean energy can reduce the particulate pollution caused by the use of traditional solid fuels; optimize the user’s home environment; reduce the probability of them suffering from cardiovascular diseases, respiratory diseases, and other diseases [27]; improve the health of family members [32]; and thus enhance the user’s well-being.

The results of models (1), (4), and (5) in Table 7 prove that household clean energy consumption significantly reduces the user’s depression, and there is a significant negative relationship between depression and WB, which means that depression plays a mediating role in the impact of CHE on WB. Specifically speaking, the consumption of clean energy by residents can save their time [26] so that they can spend more time on leisure or other things that help relieve psychological stress, reduce the possibility of their psychological abnormalities, and ultimately lead to an increase in their well-being.

Furthermore, to avoid the lack of strength of the above test [49], we verified the significance of the mediating effect using the Sobel method. The results showed that the *p*-values of Goodman-1 (Aroian), the statistic corresponding to HC and depression, were 0 and 3.142 × 10^−12^, respectively, which were less than 0.05, proving that the mediating effects of both were valid. Among them, the proportion of the mediating effect of HC and depression in the total effect was 0.501 and 0.347, respectively.

Finally, considering the limitations of the Sobel method based on the normal assumption [50], the Bootstrap method was used for further verification. The results show that none of Bootstrap’s estimated intervals contain 0, and the *p*-values are all less than 0.05. Therefore, the null hypothesis H0: ab = 0 is rejected. That is, the mediating effect of HC and depression on the influence mechanism of CHE on WB is established. In conclusion, combined with the results of the three tests, we can strongly believe that HC and depression play a mediating role in the impact mechanism of CHE on WB.

### 4.2. Moderating Effect

This part begins with “Have you interacted with friends in the last month?” As a measure of the moderating variable social contact (SC), we examined whether engaging in social activities moderated the impact of CHE on WB. The empirical results are shown in Table 8. Column (2) shows the regression results of SC and WB, implying that social contact has a significant positive effect on well-being. The coefficient of the interaction term between CHE and SC in column (3) is significant at the significance level of 1%. Combined with the regression results in columns (1) and (2), it indicates that SC plays a moderating role in the impact of CHE on WB, and SC can enhance the positive impact of CHE on WB. The above results may be explained as follows: social contact generally refers to the activities of chatting with people or carrying out topic communication, during which information about clean energy can be conveyed, and the awareness of energy use has a certain impact on energy choice [51]. Talking and exchanging ideas are conducive to the promotion of clean energy, increasing the willingness of households to use clean energy and learning from each other about related technologies and methods to enhance energy utilization. Therefore, social contact can increase clean household energy consumption.

## 5. Discussion

Modern energy use is a major issue related to future sustainable development, and its impact on well-being attracted extensive academic attention. The consumption of clean energy can bring health benefits to residents, ease the pressure of environmental pollution, improve national public health security, and play a significant role in improving the welfare of humanity. While many studies examined the effects of using clean energy, few have directly examined its impact on well-being. Via empirical analyses and mechanism tests, this study verifies the promotion effect of clean household energy consumption on residents’ well-being. This is similar to the research results of Ma [20], Maji [47], and Liu [22], etc. We further support their research conclusions. The contributions provided in this article are as follows: (1) This fills a research gap in China, a major energy consumer, and lays the groundwork for further advancement in the study of energy transition and residents’ well-being. It is the first time an extensive analysis of the impact of clean household energy consumption on residents’ well-being has been conducted from the micro-perspective of China. (2) This paper also makes a heterogeneity test of empirical results and analyzes the specific conditions of different groups of samples to make the research targeted, which provides a basis for accurately alleviating the energy problems of residents with different group characteristics. (3) It not only proposes that clean household energy consumption can improve well-being but also explores the mediating and moderating effects of health condition, depression, and social contact on the impact of clean household energy consumption on residents’ well-being. It further makes up for the lack of research attention on energy consumption as an influencing factor of well-being. 

However, our research also has limitations. On the one hand, it should be noted that we directly use life satisfaction as a proxy variable of residents’ well-being. Even though it is adopted by most researchers, it is still subjective. On the other hand, we use the micro-data of China for research. Although it can be a valuable reference for existing research, the findings may be limited to China or developing countries rather than globally. Therefore, this paper provides ideas for new research in the future: (1) Researchers can collect data covering developed and developing countries for more comprehensive research and draw conclusions with global reference values using comparative analyses. (2) Currently, academic circles focus more on the antecedents of happiness than the adverse effects of happiness. That is, further research can combine the action mechanism of residents’ happiness on energy consumption choice and comprehensively present the two-way mechanism between them.

## 6. Conclusions and Policy Recommendations

### 6.1. Conclusions

Based on CHARLS data in 2018, this study conducted empirical analysis and mechanism test on clean household energy consumption and residents’ well-being. The main conclusions of this paper are as follows: Clean household energy consumption significantly improved residents’ well-being. Furthermore, on the basis of the O-Probit model, the 2018 CGSS’s data were used to perform OLS model regression and O-Logit model regression. The results show that the relationship between clean household energy consumption and residents’ well-being is robust. Additionally, the heterogeneous impact of clean energy on the well-being of different groups was further studied, and it was found that (1) women experience a greater degree of happiness improvement when using clean energy; (2) compared with urban areas, rural users have more room for improvement in the use of clean energy, and their happiness significantly improved. Finally, we investigated the relationship between clean household energy and residents’ well-being and discovered that personal health conditions and depression mediated the relationship, and social contact moderated the influence of clean household energy on residents’ well-being.

### 6.2. Policy Recommendations 

The research conclusions in this paper have reference values for how to improve the well-being of residents in low- and middle-income countries. Finally, the following policy suggestions are put forward.

First, governments in low- and middle-income countries need to accelerate the transition and spread of clean energy and promote the adoption of clean energy by more people to maximize the benefits of clean energy.

Second, in the process of promoting clean energy, more attention should be paid to both physical and psychological feelings of users so as to improve the personal willingness of residents in using clean energy.

Third, the government should invest more efforts to protect the rights and interests of female users so that they can be liberated from the harm of traditional energy as soon as possible to safeguard gender equality.

Fourth, we suggest narrowing the gap between urban and rural endowments, improving the clean energy supply system in rural areas, and providing financial or technological support so that residents can enjoy the benefits of clean energy earlier.

## Figures and Tables

**Table 1 ijerph-19-14057-t001:** Variable’s selection and definition.

Variables’ Type	Name	Definition
Response variable	Well-Being (WB)	Self-life satisfaction, 1 = not at all satisfied; 2 = not very satisfied; 3 = somewhat satisfied; 4 = very satisfied; 5 = completely satisfied.
Explanatory variable	Cleaner household energy (CHE)	What is the main source of cooking fuel in your household? Natural-gas, marsh gas, liquefied petroleum gas and electric = clean energy = CHE = 1; coal, crop residue, and wood burning = non-clean energy = CHE = 0.
Control variables	Gender	Interviewer record the Respondent’s gender. 1 = Male; 2 = Female.
	Marriage	What is your marital status? 0 = never married, widowed, divorced and separated (don’t live together as a couple anymore); 1 = married.
	Education	What is the highest level of education you have now (not including adult education)? 1 = illiterate, did not finish primary school, home school; 2 = elementary school; 3 = middle school; 4 = high school, vocational school, associate degree; 5 = bachelor’s degree; 6 = master’s degree, 7 = doctoral degree.
	Work	Not including agricultural work, did you work for at least one hour last week? 0 = No; 1 = Yes.
	Debt	Ln(debt) = Ln (loans + credit card balance + other arrears + 1). unit: RMB
	Expenditure	Ln (annual expenditure) =Ln [(monthly expenditure) × 12 + 1]. unit: RMB
	Regions	Do you in the village or city/town? 1 = The center of city/town; 2 = Combination zone between urban and rural areas; 3 = Village; 4 = Special area.
	Medical Insurance (MI)	Have you bought medical insurance? (Include public medial insurance and private commercial medical insurance), 0 = no; 1 = yes.
	Medical Level (ML)	Are you satisfied with the quality, cost and convenience of local medical services? 1 = Very dissatisfied; 2 = Somewhat dissatisfied; 3 = Neutral; 4 = Somewhat satisfied; 5 = Very satisfied
Mediating variables	Health Condition (HC)	Would you say your health is very good, good, fair, poor or very poor? 1 = Very poor; 2 = Poor; 3 = Fair; 4 = Good; 5 = Very good.
	Depression	Assign the ten question options of depression measurement as 1–4, and finally add up according to personal options. The larger the value, the higher the depression degree.
Moderating variable	Social Contact (SC)	Have you Interacted with friends in the last month? 0 = No; 1 = Yes.
Instrumental variable	Electric Bicycle (EB)	Do you have an electric Bicycle? 0 = No; 1 = Yes.

**Table 2 ijerph-19-14057-t002:** Descriptive Statistics.

Variable	Obs	Proportion	Mean	Std. Dev.	Min	Max
WB	19,683	100.00%	3.23	0.78	1	5
WB = 1	573	2.91%				
WB = 2	1535	7.80%				
WB = 3	11,352	57.67%				
WB = 4	5295	26.90%				
WB = 5	928	4.71%				
CHE	19,683	100.00%	0.70	0.46	0	1
CHE = 0	5959	30.27%				
CHE = 1	13,724	69.73%				
Gender	19,683	100.00%	1.53	0.50	1	2
Gender = 1	9276	47.13%				
Gender = 2	10,407	52.87%				
Marriage	19,683	100.00%	0.85	0.36	0	1
Marriage = 0	2929	14.88%				
Marriage = 1	16,754	85.12%				
Education	19,683	100.00%	2.05	1.10	1	7
Education = 1	8509	43.23%				
Education = 2	4382	22.26%				
Education = 3	4295	21.82%				
Education = 4	2330	11.84%				
Education = 5	155	0.79%				
Education = 6	11	0.06%				
Education = 7	1	0.01%				
Work	19,683	100.00%	0.28	0.45	0	1
Work = 0	14,101	71.64%				
Work = 1	5582	28.36%				
Expenditure	19,683	100.00%	9.57	1.23	0	14.40
Debt	19,683	100.00%	1.63	3.82	0	15.52
ML	19,683	100.00%	3.26	1.06	1	5
ML = 1	1687	8.57%				
ML = 2	1388	7.05%				
ML = 3	9484	48.18%				
ML = 4	4351	22.11%				
ML = 5	2773	14.09%				
MI	19,683	100.00%	0.97	0.17	0	1
MI = 0	589	2.99%				
MI = 1	19,094	97.01%				
Regions	19,683	100.00%	2.52	0.81	1	4
Regions = 1	3964	20.14%				
Regions = 2	1600	8.13%				
Regions = 3	14,017	71.21%				
Regions = 4	102	0.52%				
HC	19,683	100.00%	3.05	0.99	1	5
HC = 1	1072	5.45%				
HC = 2	3675	18.67%				
HC = 3	10,378	52.73%				
HC = 4	2341	11.89%				
HC = 5	2217	11.26%				
Depression	19,683	100.00%	20.27	5.21	10	40
EB	19,683	100.00%	0.42	0.49	0	1
EB = 0	11,439	58.12%				
EB = 1	8244	41.88%				
SC	19,683	100.00%	0.34	0.47	0	1
SC = 0	13,042	66.26%				
SC = 1	6641	33.74%				

Data source: The raw data was processed using Stata v17.0 software: WB = well-being; CHE = clean household energy.

**Table 3 ijerph-19-14057-t003:** The regression results of CHE and WB.

	O-Probit (1)	O-Probit (2) Average Marginal Effect				
Variables	WB	WB = 1	WB = 2	WB = 3	WB = 1	WB = 5
CHE	0.070 *** (0.019)	−0.004 *** (0.001)	−0.008 *** (0.002)	−0.012 *** (0.003)	0.017 *** (0.005)	0.007 *** (0.002)
Gender	−0.091 *** (0.017)					
Marriage	0.129 *** (0.023)					
Education	−0.009 (0.008)					
Work	0.015 (0.019)					
Expenditure	0.026 *** (0.007)					
Debt	−0.026 *** (0.002)					
MI	0.156 *** (0.046)					
ML	0.236 *** (0.008)					
Regions	−0.015 (0.011)					
Observations	19,683	19,683	19,683	19,683	19,683	19,683

Note: Robust standard errors in parentheses, *** *p* < 0.01. WB = well-being; 1 = not at all satisfied; 2 = not very satisfied; 3 = somewhat satisfied; 4 = very satisfied; 5 = completely satisfied. CHE = clean household energy; 1 = clean energy; 0 = non-clean energy. MI = medical insurance. ML = medical level.

**Table 4 ijerph-19-14057-t004:** Robustness test results for replacement sample data and models.

	CHARLS			CGSS_2018		
	O-Probit (1)	OLS (2)	O-Logit (3)	O-Probit (4)	OLS (5)	O-Logit (6)
Variables	WB	WB	WB	WB	WB	WB
CHE	0.070 ***	0.049 ***	0.129 ***	0.087 **	0.082 ***	0.137 *
	(0.019)	(0.013)	(0.033)	(0.040)	(0.029)	(0.071)
	Average marginal effect					
WB = 1	−0.004 ***		−0.004 ***	−0.003 **		−0.002 *
	(0.001)		(0.001)	(0.001)		(0.001)
WB = 2	−0.008 ***		−0.008 ***	−0.010 **		−0.008 *
	(0.002)		(0.002)	(0.004)		(0.004)
WB = 3	−0.012 ***		−0.143 ***	−0.013 **		−0.013 *
	(0.003)		(0.004)	(0.006)		(0.007)
WB = 4	0.017 ***		0.021 ***	0.002 *		0.003 *
	(0.005)		(0.005)	(0.001)		(0.002)
WB = 5	0.007 ***		0.006 ***	0.023 **		0.020 *
	(0.002)		(0.001)	(0.010)		(0.010)
CV	Control	Control	Control	Control	Control	Control
Observations	19683	19,683	19,683	3603	3603	3603

Note: Robust standard errors in parentheses, *** *p* < 0.01, ** *p* < 0.05, * *p* < 0.1. WB = well-being; 1 = not at all satisfied; 2 = not very satisfied; 3 = somewhat satisfied; 4 = very satisfied; 5 = completely satisfied. CHE = clean household energy; 1 = clean energy; 0 = non-clean energy. CGSS_2018 = the 2018 Chinese General Social Survey Data. CV = control variables.

**Table 5 ijerph-19-14057-t005:** The results of the IV-O-Probit model for endogenous issues.

	TSLS		CMP Estimation Method	
	First-Stage	Second-Stage	IV-O-Probit	
	(1)	(2)	(3)	(4)
Variables	CHE	WB	CHE	WB
CHE		0.638 ***		0.313 ***
		(0.121)		(0.065)
EB	0.096 ***		0.034 ***	
	(0.006)		(0.019)	
F-statistics	533.31			
atanhrho_12 (*p*)				0.000
CV	Control	Control	Control	Control
Observations	19,683	19,683	19,683	19,683

Note: Robust standard errors in parentheses, *** *p* < 0.01. WB = well-being; 1 = not at all satisfied; 2 = not very satisfied; 3 = somewhat satisfied; 4 = very satisfied; 5 = completely satisfied. CHE = clean household energy; 1 = clean energy; 0 = non-clean energy. EB = Do you have electric Bicycle? 0 = No; 1 = Yes; CV = control variables.

**Table 6 ijerph-19-14057-t006:** The results of heterogeneity analysis.

	O-Probit	O-Probit	O-Probit	O-Probit	O-Probit
	(1)	(2)	(3)	(4)	(5)
	male	female	urban	urban-rural combination	rural
Variables	WB	WB	WB	WB	WB
CHE	0.0699 **	0.0716 **	0.0426	−0.0205	0.0706 ***
	(0.0278)	(0.0262)	(0.0852)	(0.0961)	(0.0198)
CV	Control	Control	Control	Control	Control
Observations	9276	10,407	3964	1600	14,017

Note: Robust standard errors in parentheses, *** *p* < 0.01, ** *p* < 0.05. WB = well-being; 1 = not at all satisfied; 2 = not very satisfied; 3 = somewhat satisfied; 4 = very satisfied; 5 = completely satisfied; CHE = clean household energy; 1 = clean energy; 0 = non-clean energy; Gender = interviewer record the respondent’s gender. 1 = Male; 2 = Female. Regions = Do you in the village or city/town? 1 = the center of city/town; 2= combination zone between urban and rural areas; 3 = village; CV= control variables.

**Table 7 ijerph-19-14057-t007:** The results of mediating effect test for CHE and WB:HC and Depression.

	O-Probit (1)	O-Probit (2)	O-Probit (3)	O-Probit (4)	O-Probit (5)
Variables	WB	HC	WB	Depression	WB
CHE	0.070 ***	0.159 ***	0.035 *	−0.114 ***	0.046 **
	(0.019)	(0.018)	(0.019)	(0.018)	(0.019)
HC			0.268 ***		
			(0.009)		
Depression					−0.040 ***
					(0.002)
Goodman-1(Aroian) (*p*)		0.000 < 0.05		3.142 × 10^−12^ < 0.05	
		Indirect effect/Total effect = 0.501		Indirect effect/Total effect = 0.347	
Bootstrap (500)		Indirect effect (*p* = 0.000 < 0.05)		Indirect effect (*p* = 0.000 < 0.05)	
		Direct effect (*p* = 0.039 < 0.05)		Direct effect (*p* = 0.010 < 0.05)	
CV	Control	Control	Control	Control	Control
Observations	19,683	19,683	19,683	19,683	19,683

Note: Robust standard errors in parentheses, *** *p* < 0.01, ** *p* < 0.05, * *p* < 0.1. WB = well-being; 1 = not at all satisfied; 2 = not very satisfied; 3 = somewhat satisfied; 4 = very satisfied; 5 = completely satisfied. CHE = clean household energy; 1 = clean energy; 0 = non-clean energy. HC = Would you say your health is very good, good, fair, poor, or very poor? 1 = very poor; 2 = poor; 3 = fair; 4 = good; 5= very good. Depression= Assign the ten question options of depression measurement as 1–4 and finally add them up according to personal options. The larger the value, the higher the depression degree. CV = control variables.

**Table 8 ijerph-19-14057-t008:** The results of the moderation effect test for CHE and WB: SC.

	O-Probit (1)	O-Probit (2)	O-Probit (3)
Variables	WB	WB	WB
CHE	0.070 ***		
	(0.019)		
SC		0.067 ***	
		(0.017)	
CHE × SC			0.078 ***
			(0.019)
CV	Control	Control	Control
Observations	19,683	19,683	19,683

Note: Robust standard errors in parentheses, *** *p* < 0.01. WB = well-being; 1 = not at all satisfied; 2 = not very satisfied; 3 = somewhat satisfied; 4 = very satisfied; 5 = completely satisfied. CHE = clean household energy; 1 = clean energy; 0 = non-clean energy. SC = social contact = Have you Interacted with friends in the last month? 0 = No; 1 = Yes; CV = control variables.

## Data Availability

The datasets used or analyzed during the current research are available from the corresponding author upon reasonable request.

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
