# Peer review of "Clean Household Energy Consumption and Residents’ Well-Being: Empirical Analysis and Mechanism Test"

_ijerph, 2022, doi:10.3390/ijerph192114057_

Round 1

Reviewer 1 Report

The article addresses a very interesting topic. The article is well structured, the ideas are presented in a logical, concise order. The statements are supported by statistical data, that are analysed in a proper way by the authors.

The article has the potential to be published, which is why I recommend some revisions

1.      Authors must explain better this connection ”Lack of access to clean cooking  fuel is one of the key causes of energy poverty….line 38 page 1.

2.      In introduction, authors can mentions some of the achievements of the European Union to reduce energy poverty, considering the intense concerns of the member countries in this regard. https://energy.ec.europa.eu/topics/markets-and-consumers/energy-consumer-rights/energy-poverty-eu_en

3.      The issue of energy poverty must be better integrated in the context of two somewhat contradictory processes considering the current geopolitical context, namely energy transition and energy security

4.      Authors must insert a generous literature review section in order to present the results of the main studies and to identiy the research gap.

5.      The authors must better justify the choice of the method usedThe discussion section must be extended. The authors must present their results in the context of similar studies that confirm or refute their conclusions

6.      In the conclusions section, the authors must present the limits of research and future directions for their research

7.      The references section must be extended considerably and the results of additional papers must be presented in LR section or discussion sections. Several recommendations are

Bouzarovski, S., Thomson, H., & Cornelis, M. (2021). Confronting energy poverty in Europe: A research and policy agenda. Energies14(4), 858.

Neacsa, A., Panait, M., Muresan, J. D., & Voica, M. C. (2020). Energy poverty in European Union: Assessment difficulties, effects on the quality of life, mitigation measures. some evidences from Romania. Sustainability12(10), 4036.

Polimeni, J. M., Simionescu, M., & Iorgulescu, R. I. (2022). Energy Poverty and Personal Health in the EU. International Journal of Environmental Research and Public Health19(18), 11459.

Ulucak, R., Sari, R., Erdogan, S., & Alexandre Castanho, R. (2021). Bibliometric Literature Analysis of a Multi-Dimensional Sustainable Development Issue: Energy Poverty. Sustainability13(17), 9780.

.

Reviewer 2 Report

This study is important. However, there are still some major issues that must be resolved. Please see below specific comments to help improve the paper. 

  1. Please check the abstract again. The abstract must briefly summarize the problem, state the research aim, and specific objectives, briefly describe the methodology, and present the key findings and implications of the key findings. This must be done.

2.       Page 2 line 64, please check the reference Krishnapriya, P.P. It’s not well referenced.

3.       The introduction is well written. The problem is well-stated and discussed. The research gaps are also well-stated. However, please state the research aim and specific objectives at the end of the introduction before you introduce readers to what to expect in the entire paper.

4.       Please check the sub-topic for Section 2.1 well. ‘Data’ does not seem to depict the contents of the sub-topic 2.1. I see a questionnaire in there. Are the authors referring to Data Collection Technique?

5.       Lines 103-105 depict that this study used a secondary data source. I think lines 105-106 defeat this purpose. Please delete the sentence “The content of the questionnaire includes basic personal information of respondents……”. This tells readers that the authors collected the data themselves which was not the case in this paper. Unless there is more to this.

6.       How were the mean filling and the elimination done? Please provide some further clarity on the two under Section 2.1.

7.       Please check Section 2.2. The information provided there seems to contradict Section 2.1. From Section 2.2, the paper seems to be describing the contents of a particular questionnaire which means the paper sought primary information from the respondents. It is not clear whether there was a combination of both primary and secondary data sources for this study. If this was the case, there is a need for clarity. At what point was the secondary data source used and at what point was the primary data source used?

8.       Who constituted the population of this study and in which parts of China were they sourced? Again what was the sample size for this study and which sampling technique was used to select the respondents?

9.       Section 2.3 seems to be presenting some results. If that section is for the results it will be good to title it as such.

10.   How were the data analyzed? The software is stated, however, the statistical tools used are not described.

11.   Please check lines 166-168 for potential errors.

12.   What is Section 3.1 trying to achieve? And which specific objective is it addressing? I would suggest to the authors present the key findings with clear objectives.

13.   Please separate the discussion section from the conclusion and recommendations.

14.   The results have not been discussed at all. Please provide a separate section that recaps the study aim and specific objectives and that discusses the results under the specific objectives. Please ensure that the findings are critically compared with the related literature in the area.

15.   In the concluding section, it will be good to also recap the gaps identified earlier on in the introduction and discuss how this study has bridged these gaps. 

Round 2

Reviewer 1 Report

The authors improved the manuscript considerably and took into account the recommendations made, which means that it can be published, but small adjustments are still needed.

1. The discussion section should be expanded, because the authors have added some ideas. The mentions of limits and future research directions should be moved to the conclusions section.

2. The authors must be careful with the terms used, I think it is a  translation error  ....MICROSCOPIC  data (line 452)....probably the intention is to use the sintagma DATA for  microeconomic level....

Thanks to the authors and editors for the opportunity to review this work. Good luck to everyone.

Reviewer 2 Report

I wish to thank the authors for taking the time to address the majority of my comments. Notwithstanding, there are still other comments that were made previously but were not addressed. Please see them below.

1.       In Line 18, what does (IV) mean? Must it be there?

2.       Page 2 line 82, please check the reference Krishnapriya, P.P. It’s not well referenced.

3.       Please check the sub-topic for Section 2.1 well. ‘Data’ does not seem to depict the contents of the sub-topic 2.1. I see a questionnaire in there. Are the authors referring to Data Collection Technique?

4.       Who constituted the population of this study and in which parts of China were they sourced? Again what was the sample size for this study and which sampling technique was used to select the respondents?

5.       Section 2.3 seems to be presenting some results. If that section is for the results it will be good to title it as such.

6.       How were the data analyzed? The software is stated, however, the statistical tools used are not described.

7.       What is Section 3.1 trying to achieve? And which specific objective is it addressing? I would suggest to the authors present the key findings with clear objectives.
